# Magnesium Hydroxide Nanoparticles Kill Exponentially Growing and Persister *Escherichia coli* Cells by Causing Physical Damage

**DOI:** 10.3390/nano11061584

**Published:** 2021-06-16

**Authors:** Yohei Nakamura, Kaede Okita, Daisuke Kudo, Dao Nguyen Duy Phuong, Yoshihito Iwamoto, Yoshie Yoshioka, Wataru Ariyoshi, Ryota Yamasaki

**Affiliations:** 1Department of Health Promotion, Division of Infections and Molecular Biology, Kyushu Dental University, Kitakyushu 803-8580, Fukuoka, Japan; r19nakamura@fa.kyu-dent.ac.jp (Y.N.); r19okita@fa.kyu-dent.ac.jp (K.O.); r16yoshioka@fa.kyu-dent.ac.jp (Y.Y.); arikichi@kyu-dent.ac.jp (W.A.); 2Kyowa Chemical Industry Co., Ltd., Hayashida-cho, Sakaide 762-0012, Kagawa, Japan; daisukekudo@kyowa-chem.co.jp (D.K.); daonguyenduyphuong@kyowa-chem.co.jp (D.N.D.P.); yoshihitoiwamoto@kyowa-chem.co.jp (Y.I.)

**Keywords:** magnesium hydroxide nanoparticle, *Escherichia coli*, sterilization, persister

## Abstract

Magnesium hydroxide nanoparticles are widely used in medicinal and hygiene products because of their low toxicity, environment-friendliness, and low cost. Here, we studied the effects of three different sizes of magnesium hydroxide nanoparticles on antibacterial activity: NM80, NM300, and NM700. NM80 (D_50_ = 75.2 nm) showed a higher bactericidal effect against *Escherichia coli* than larger nanoparticles (D_50_ = 328 nm (NM300) or 726 nm (NM700)). Moreover, NM80 showed a high bactericidal effect against not only exponential cells but also persister cells, which are difficult to eliminate owing to their high tolerance to antibiotics. NM80 eliminated strains in which magnesium-transport genes were knocked out and exhibited a bactericidal effect similar to that observed in the wild-type strain. The bactericidal action involved physical cell damage, as confirmed using scanning electron microscopy, which showed that *E. coli* cells treated with NM80 were directly injured.

## 1. Introduction

Magnesium hydroxide is globally used as a dietary supplement for magnesium, an essential mineral, as well as a pH adjuster and color stabilizer for foods, because of its low cost and minimum impact on the environment and the human body. Additionally, magnesium hydroxide is used in various products, such as antacids [1], laxatives [2], fertilizers [3], PVC materials (as stabilizers) [4], and resins (as flame retardants) [5], because of its favorable chemical properties. Magnesium hydroxide nanoparticles have medicinal properties; they are used in glaucoma treatment [6]; in biomaterials, to create implants that induce low inflammation [7]; and in antibacterial materials [8,9]. In recent years, “antimicrobial nanoparticles” have been widely studied for their high sterilizing ability, safety, and practicality. In addition to magnesium hydroxide, titanium dioxide [10], zinc oxide [11], and silver oxide [12] have all been reported as antimicrobial nanoparticles. These metal oxide nanoparticles are mainly used for sterilization because of their photocatalytic properties or their ability to generate reactive oxygen species.

In this study, the antibacterial activities of magnesium hydroxide nanoparticles of different sizes were examined. Their bactericidal effects have already been reported [9]; however, this report describes the elimination of bacterial persister cells using magnesium hydroxide nanoparticles and discusses the plausible sterilization mechanisms. Specifically, we examined the effect of magnesium hydroxide nanoparticles on *Escherichia coli* persister cells. Persister cells are a subpopulation of cells that appear upon stress exposure [13]. Persister cells were first discovered by Hobby et al. [14], and to date, bacterial persisters of various species, such as *E. coli* [15,16,17,18], *Pseudomonas aeruginosa* [19], and *Salmonella* [20,21], have been studied. Pathogenic bacteria can also form persisters, allowing them to avoid antibiotics and other drugs, and repopulate when the stress is relieved. Thus, persisters are detrimental to medicine, the food industry, and the environment.

In a previous study, a method to artificially induce persister cells with high probability (>80%) was established [22,23], and many research groups have used this method to study persister cells [24,25,26,27,28,29]. Thymine has been reported to improve the antibacterial activity of some antibiotics against persister cells [30]. Moreover, mitomycin C [31] and copper [32] can kill *E. coli* persister cells. Here, we revealed the bactericidal effect of magnesium hydroxide nanoparticles on *E. coli* exponential and persister cells, and their underlying mechanisms of action.

## 2. Materials and Methods

### 2.1. Preparation and Characterization of Magnesium Hydroxide Nanoparticles

The magnesium hydroxide nanoparticles were synthesized as described in the following steps. Magnesium chloride and sodium hydroxide were allowed to react continuously at 500 rpm at 20 °C. In the case of NM80, 1000 mL of 0.2 mol/L magnesium chloride and 640 mL of 0.5 mol/L sodium hydroxide were used for the reaction. For NM300, 500 mL of 2.0 mol/L magnesium chloride and 150 mL of 12.0 mol/L sodium hydroxide were used for the reaction. For NM700, 2825 mL of 1.8 mol/L magnesium chloride and 2553 mL of 2.2 mol/L sodium hydroxide were used for the reaction. The obtained magnesium hydroxide was filtered, washed with ion-exchange water, and then stirred with a homogenizer at 5000 rpm for 30 min. Magnesium hydroxide nanoparticles were then allowed to mature for 4 h under stirring at 500 rpm while maintaining the temperature at 50 °C. In the case of NM700, the nanoparticles were matured for 2 h under stirring at 620 rpm while maintaining the temperature at 170 °C. The magnesium hydroxide nanoparticles were characterized using scanning electron microscopy (SEM; S-4300, HITACHI, Tokyo, Japan), dynamic light scattering (DLS; ELSZ-2000, Otsuka Electronics Co., Ltd., Osaka, Japan), laser diffraction particle size analysis (MT3000II, MicrotracBEL Corp., Osaka, Japan), inductively coupled plasma optical emission spectrometry (ICP-OES; SPS3500DD, HITACHI, Tokyo, Japan), pH measurement (LAQUAact D-71, HORIBA, Kyoto, Japan), and X-ray diffraction (XRD; Empyrean, PANalytical, Almelo, The Netherlands). The XRD patterns of the nanoparticles were recorded using a Cu K_α_ radiation (40 kV, 45 mA) source in continuous scanning mode. The data were collected over a 2θ range of 2° to 70° with 0.0263° scan steps and a collection time of 7.14 s per step for NM80. For NM300 and NM700, the data were collected in steps of 0.0066° with a collection time of 13.77 s per step. The results obtained were cross-referenced to the ICDD database (No. 01-083-0114) for analysis. The average crystallite size was determined from the broadness of the peaks using the Scherrer equation [33].

### 2.2. Bacteria Cultivation

The bacterial strains and plasmids used in this study are listed in Table 1. *Escherichia coli* K-12 BW25113 and its isogenic mutants [34] were used in this study. Wild type was grown in lysogeny broth (LB; Difco Laboratories, Detroit, MI, USA) [35], and mutants were grown in LB with 50 mg/mL kanamycin (FUJIFILM Wako Pure Chemical Corporation, Osaka, Japan) at 37 °C.

### 2.3. Bactericidal Effect of NM80, NM300, and NM700

A single colony of *E. coli* was inoculated into LB and incubated at 37 °C. An overnight culture was inoculated into fresh LB at 1/100 dilution and incubated to a turbidity of 0.6 at 600 nm. The culture was centrifuged at 5000× *g* for 2 min and washed twice with 1× PBS [36]. The bacterial pellet was re-suspended in 1× PBS containing magnesium hydroxide nanoparticles (NM80, NM300, or NM700) at a concentration of 10–500 mg/L and incubated at 37 °C. After treatment with magnesium hydroxide nanoparticles, the samples were diluted by a 10-fold serial dilution and plated on LB agar to measure the number of colony-forming units. Preparation of rifampicin-induced *E. coli* persister cells was performed as described previously [22,23]. Briefly, most of the *E. coli* formed persisters (>80%) upon treatment with rifampicin (100 µg/mL) for 30 min. Non-persister cells were removed by ampicillin treatment. The bactericidal effects of the magnesium hydroxide nanoparticles were also examined using the same method.

### 2.4. SEM Analysis of E. coli

SEM samples were prepared in accordance with a previous study [37]. The *E. coli* cells treated with NM80 were plated onto a mixed cellulose ester membrane (Merck Millipore, Billerica, MA, USA). The bacterial cells were fixed using 2% glutaraldehyde fixative solution and washed twice with 0.1 M phosphate buffer (pH 7.4), then treated with 50%, 70%, 90%, 95%, and 100% (3 times) acetone to dehydrate the samples. Acetone was discarded, t-butanol was added, and the cells were frozen at −30 °C. The frozen bacterial samples were lyophilized, and a thin layer of platinum was deposited over the samples prior to imaging. The prepared bacterial samples were visualized using SEM.

## 3. Results

### 3.1. Characterization of Magnesium Hydroxide Nanoparticles

Magnesium hydroxide nanoparticles (NM80, NM300, and NM700) were imaged using SEM. As observed in the micrographs, NM80 exhibited the smallest size of nanoparticles (50–100 nm), NM300 exhibited the intermediate size (200–400 nm), and NM700 exhibited the largest size (400–1000 nm) (Figure 1A). From the results of the DLS and the laser diffraction particle size analyzer, the median particle diameter (D_50_) of NM80, NM300, and NM700 was found to be 75.2 nm, 328 nm, and 726 nm, respectively (Figure 1B, Table 2). The XRD results also showed that the peak width became narrower as the particle size increased, indicating an increase in the grain size as the particle size increased (Figure 1C). Assuming the X-ray wavelength to be λ = 0.154 nm and the Scherrer constant K = 0.9, 2θ = 18.55 (deg) (θ = 0.16 rad) and B = 0.0144 rad were obtained from the measurement, and the crystallite size D of the planes (101) in NM80 were determined to be 16.1 nm from the Scherrer equation (D = Kλ/Bcosθ). Similarly, the crystallite sizes of planes (101) in NM300 and NM700 were determined to be 42.3 nm and 51.5 nm, respectively. The XRD data are shown in Appendix A. Thus, the magnesium hydroxide nanoparticles appeared as small flakes consisting of dense crystallites, which were 4 to 14 times smaller than the respective particle sizes. The amount of magnesium ions in the solvents were approximately 5 ppm, and the pH was approximately 10.2, which is independent of the nanoparticle sizes (Table 2).

### 3.2. Bactericidal Effects of NM80, NM300, and NM700 on E. coli

Figure 2 shows the concentration-dependent bactericidal effect of NM80, measured over a concentration range of 10 to 500 mg/L. The bactericidal effect increased with increasing concentration of NM80. At a concentration of 500 mg/L, *E. coli* cells were eliminated after an 18 h incubation period. At a concentration of 100 mg/L, NM80 showed a slight bactericidal effect after 18 h of treatment (Figure 2). Comparing the activities of NM300 and NM700 with that of NM80 revealed a decreased antibacterial activity with increasing particle size (Figure 3). MgSO_4_ was also used at a concentration of 500 mg/L; however, there was no effect on the bacterial cells after 18 h treatment (Appendix A). In addition, the bactericidal effects of different concentrations of NM300 and NM700 (10, 50, and 100 mg/L) were also analyzed. Lower concentrations of magnesium hydroxide nanoparticles (<100 mg/L) exhibited no bactericidal effect, similar to NM80 (Appendix A).

### 3.3. Magnesium Ions Do Not Affect Bacterial Death

Magnesium ions tended to leach out of the nanoparticles. Therefore, the effect of the leached out Mg^2+^ on bacterial growth was investigated using the knockout strain *E. coli* BW25113, which lacks genes involved in magnesium ion transport (Δ*corA*, Δ*mgtA*, Δ*pitA*, Δ*pitB*, and Δ*yifB*). CorA is a magnesium-transport protein that mediates the influx of magnesium ions [38]. MgtA is a magnesium-transporting ATPase that mediates magnesium influx into the cytosol [39]. In addition, CorA and MgtA (also termed as CorB) are related to the efflux of magnesium ions [40]. PitA and PitB are H^+^ symporters that depend on the presence of Mg^2+^ [41]. YifB is putatively a magnesium chelatase belonging to the Helix 2 insert clade of the AAA+ ATPases and functions as a molecular chaperone and ATPase subunit of proteases, helicases, or nucleic-acid-stimulated ATPases [42]. Similar to the wild-type strain, all *E. coli* mutants were killed by NM80 (Figure 4). Therefore, our results indicate that magnesium ion uptake, discharge via magnesium transporters, or other properties of magnesium ions have no effect on antibacterial activity. In addition, ICP measurements showed that the amounts of magnesium ions in the NM80, NM300, and NM700 solutions were almost equal (Table 2). These results indicated that sterilization was not due to the chemical action of the magnesium ions.

### 3.4. Smaller Magnesium Hydroxide Nanoparticles Physically Injure Bacteria and Kill Them

Next, physical damages induced by magnesium hydroxide nanoparticles were investigated (Figure 3 and Figure 4). *E. coli* cells treated with NM80, NM300, and NM700 for 18 h were observed using SEM. Although the PBS-treated *E. coli* retained their shape (there were no injuries) (Figure 5A), NM80-treated *E. coli* did not retain their original form (Figure 5B, left). When a portion of the image was magnified, many scratches of several tens of nanometers in size were observed (Figure 5B, right). The median particle diameter (D_50_) of NM80 was 75.2 nm (Figure 1, Table 2); therefore, it can be concluded that NM80 physically injured the cell membrane, causing it to become lethal. Some of the NM300-treated cells showed damage but mostly retained their original shape when compared with NM80-treated cells (Figure 5C). In the case of the NM700 treatment, the treated cells retained their original shape and were largely unaltered (Figure 5D). The change in the shape of the bacteria seen in the SEM images correlates with the bactericidal curve shown in Figure 3, indicating that the bacteria were killed by physical injury.

### 3.5. NM80 Sterilizes even Persister Cells with High Efficiency

To examine the effect of NM80 on persister cells, which are highly tolerant to various stresses, the bactericidal effect of NM80 on rifampicin-induced *E. coli* persister was evaluated. NM80 showed a high disinfection effect against persister cells, similar to exponential cells (Figure 6). The SEM images in Figure 5 show that the bactericidal effect of NM80 is caused by physical injury. Therefore, even persisters, which are resistant to chemical and environmental stresses, can be easily sterilized by NM80.

## 4. Discussion

The results show that the bactericidal effect of magnesium hydroxide nanoparticles is not due to a chemical alteration in cells, but rather due to physical damage. It is also clear that there is no effect of pH on this process, as the pH of each sample was almost the same at approximately 10 (Table 2), and *E. coli* was not sterilized in pH 10 PBS solution (Appendix A). However, the bactericidal effects of magnesium hydroxide nanoparticles were found to be influenced mainly by morphological features of the nanoparticles, especially their size. As shown in the SEM image in Figure 5, the smallest of the three types of nanoparticles studied produced the most lethal wounds on *E. coli* cells. As the thickness of the nanoparticles was approximately 25 nm (Appendix A), the contour part of the nanoparticles was considered important for inducing bacterial death. For the same total area occupied by the nanoparticle, NM80 exhibited a higher bactericidal effect than NM700 because the total length of the contour increases as the particle size decreases (Figure 7A). Moreover, the size of NM700 did not vary significantly from that of *E. coli*, whereas that of NM80 was considerably smaller. Therefore, it is assumed that the killing ability was improved by particles of smaller size (Figure 7B). To clarify the ratio relationship between the size of the nanoparticles and the size of the sterilization target, it is necessary to verify the results with larger microorganisms (e.g., fungi and protozoa).

In a previous study, it was reported that at a concentration of over 1.25 M, MgSO_4_ can kill *E. coli* [43]. The concentration of MgSO_4_ used in this study was 0.004 M (500 mg/L) which, naturally, was insufficient to kill the cells (Appendix A). However, at the same concentration, magnesium hydroxide nanoparticles (especially NM80) could kill *E. coli* cells completely after 18 h of treatment (Figure 3). In addition, NM80 could kill *E. coli* persister cells at a concentration of 500 mg/L (Figure 6). Previously, CuSO_4_ was demonstrated to have a bactericidal effect on *E. coli* persister cells [32]. It was reported that CuSO_4_ reduced the number of *E. coli* persister cells by 1000-fold after 18 h of incubation at 960 mg/L concentration. However, NM80 eliminated the *E. coli* persister cells after 18 h of treatment at a much lower concentration (500 mg/L). This concentration has been proven to have no effect on the human body even when administered orally [44]. Thus, it can be used in various applications, such as the food industry and in medicine, and it is environmentally friendly.

## 5. Conclusions

This study reports that magnesium hydroxide nanoparticles (NM80) exhibited a high bactericidal effect toward *E. coli*. Importantly, NM80 eliminates not only *E. coli* exponential cells but also persister cells. The mechanism of the bactericidal effect is primarily physical injury, as SEM observations revealed numerous scratches on the cell surface caused by the nanoparticles. Although the present study shows the bactericidal effect of magnesium hydroxide nanoparticles on *E. coli* and its persister cells, it can be expected to show similar bactericidal effects on various other bacteria. In addition, as magnesium hydroxide is inexpensive, environmentally friendly, and has little impact on the human body, it is likely to have a wide range of applications as an antibacterial drug in medicine and as a general disinfectant in the environment.

## Figures and Tables

**Figure 1 nanomaterials-11-01584-f001:**
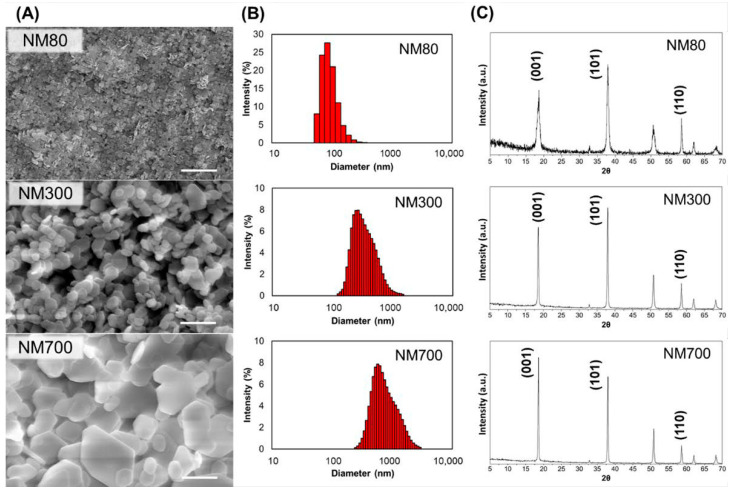
Characterization of NM80, NM300, and NM700 using SEM, DLS, and XRD. (**A**) SEM images of NM80, NM300, and NM700 from the top. Scale bars indicate 1 µm (×20,000). (**B**) Nanoparticle size measurement using DLS or laser diffraction particle size analyzer. (**C**) XRD patterns and important directions of magnesium hydroxide nanoparticles are indicated ((001), (101), and (110) from the left).

**Figure 2 nanomaterials-11-01584-f002:**
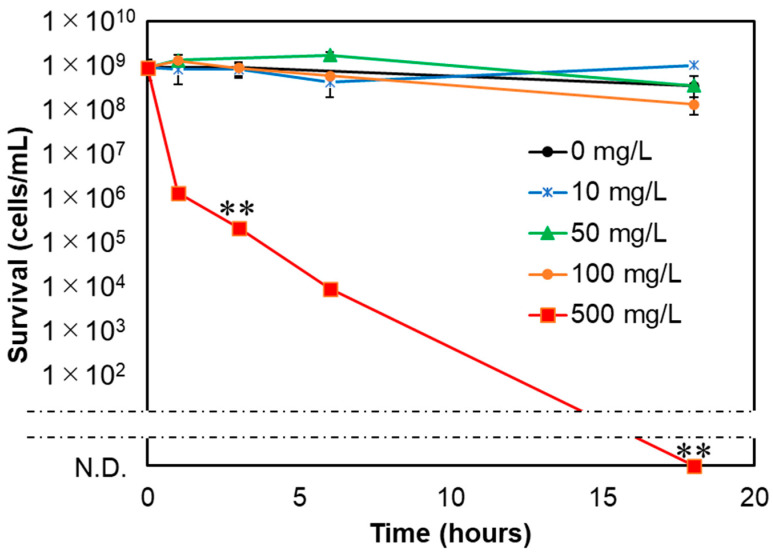
Bactericidal effect of NM80 at concentration dependence. Different concentrations of NM80 at 500 mg/L (red), 100 mg/L (orange), 50 mg/L (green), and 10 mg/L (blue) mg/L were examined to assess the ability of NM80 to eliminate *E. coli*. N.D. means “not detected,” i.e., completely sterilized. Student’s *t*-tests were used to compare the bactericidal effects of different NM80 concentrations relative to 0 mg/L NM80 (** indicates a *p*-value < 0.01).

**Figure 3 nanomaterials-11-01584-f003:**
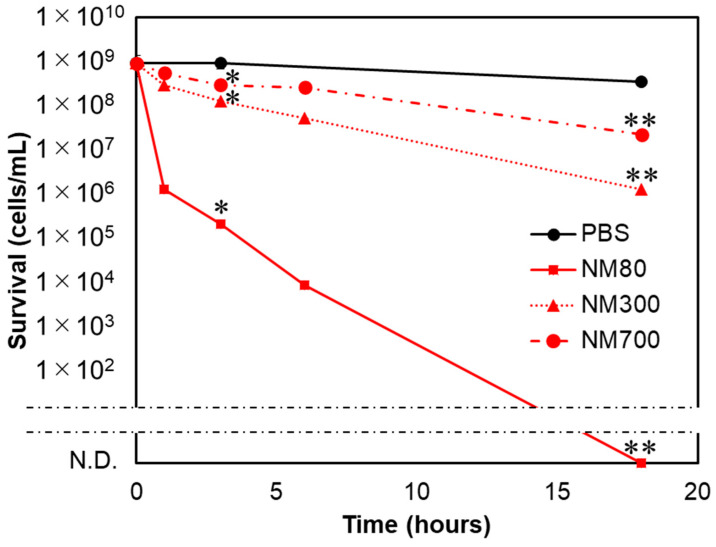
Size-dependent bactericidal effects of magnesium hydroxide nanoparticles. Different sizes of the magnesium hydroxide nanoparticles (NM80, NM300, and NM700) were compared for their bactericidal effects against *E. coli*. The concentration of each nanoparticle used was 500 mg/L. The characteristics of each nanoparticle are indicated in Figure 1 and Table 2. The black line indicates the PBS treatment as a control. N.D. means “not detected.” Student’s *t*-tests were used to compare bactericidal effects of each sample relative to that of PBS (* indicates a *p*-value < 0.05 and ** indicates a *p*-value < 0.01).

**Figure 4 nanomaterials-11-01584-f004:**
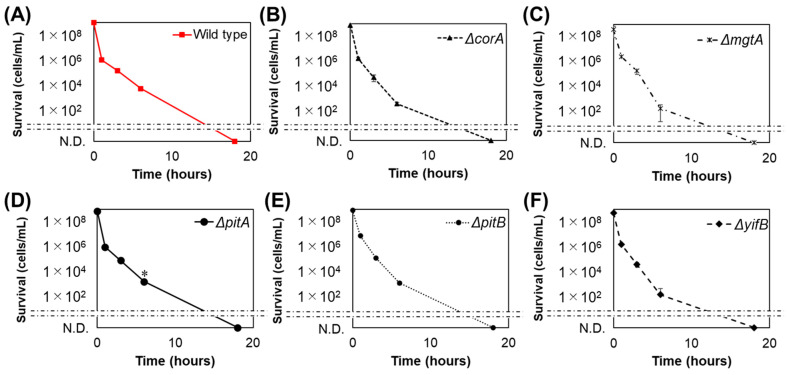
Bactericidal effect of NM80 against the magnesium-transport-related knockout strains: (**A**) wild type, (B) Δ*corA*, (**C**) Δ*mgtA*, (**D**) Δ*pitA*, (**E**) Δ*pitB*, and (**F**) Δ*yifB*. NM80 concentration was 500 mg/L. N.D. means “not detected.” Student’s *t*-tests were used to compare the wild type with each mutant (* indicates a *p*-value < 0.05).

**Figure 5 nanomaterials-11-01584-f005:**
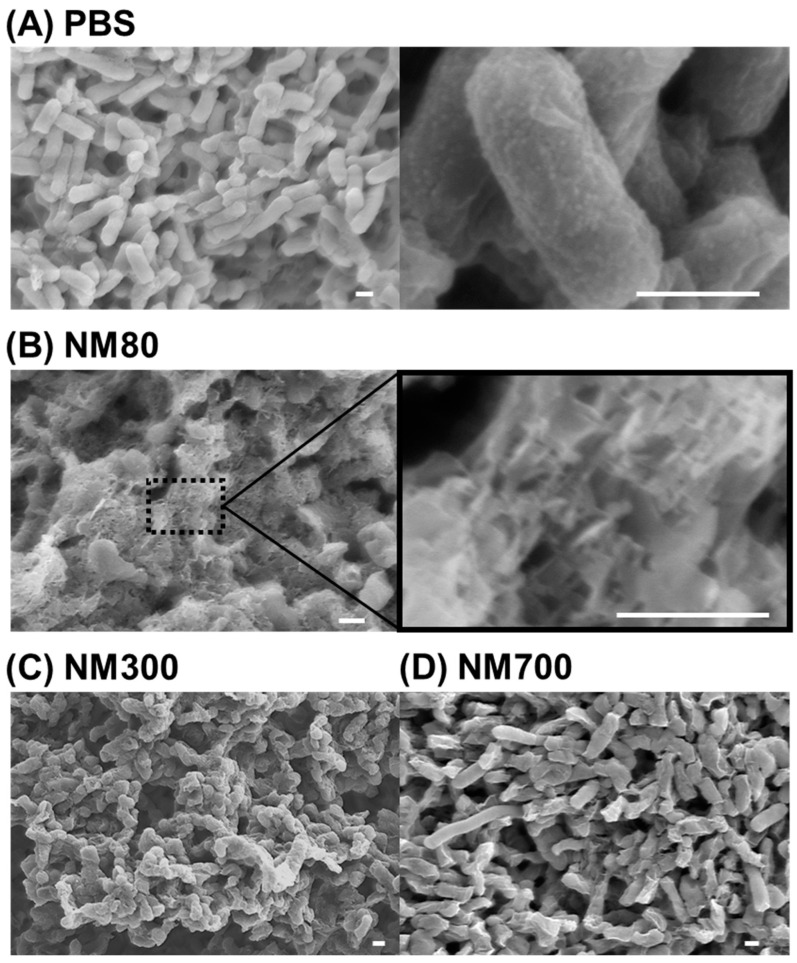
SEM images of *E. coli* treated by NM80, NM300, and NM700. (**A**) PBS-treated *E. coli* as a control. Scale bar indicates 2 µm (×10,000). Right image is a magnified image. Scale bar indicates 500 nm (×80,000). (**B**) NM80-treated *E. coli*. Scale bar indicates 1 µm (×15,000). Right image is a magnified image. Scale bar indicates 500 nm (×100,000). (**C**) NM300-treated *E. coli*. Scale bar indicates 2 µm (×7000). (**D**) NM700-treated *E. coli*. Scale bar indicates 2 µm (×9000). All samples were treated with the appropriate magnesium hydroxide nanoparticles for 18 h and prepared for SEM observation following the protocol given in the Methods section.

**Figure 6 nanomaterials-11-01584-f006:**
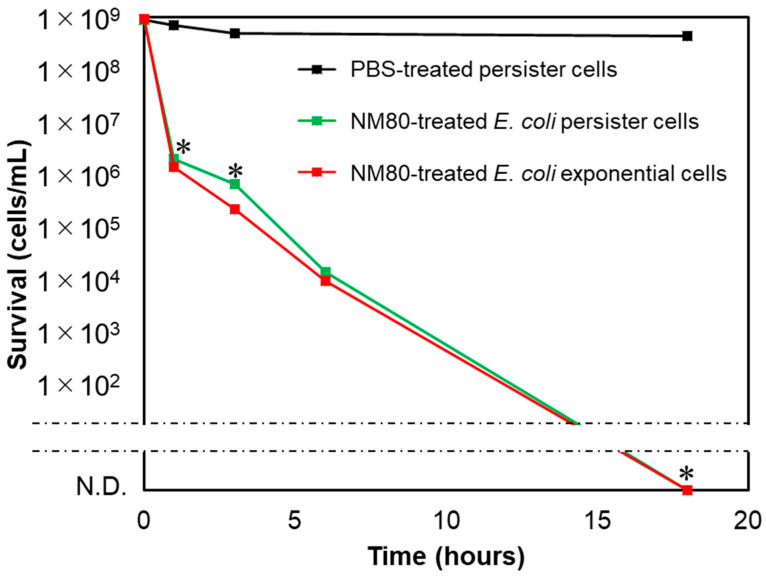
Bactericidal effect of NM80 against *E. coli* persister cells. The red line is the NM80-treated *E. coli* exponential cells, and the green line indicates NM80-treated *E. coli* persister cells. The black line indicates the PBS-treated *E. coli* persister cells as a control. N.D. means “not detected.” Student’s *t*-tests were used to compare the bactericidal effects of nanoparticle-treated samples relative to PBS-treated samples (persister) (* indicates a *p*-value < 0.05).

**Figure 7 nanomaterials-11-01584-f007:**
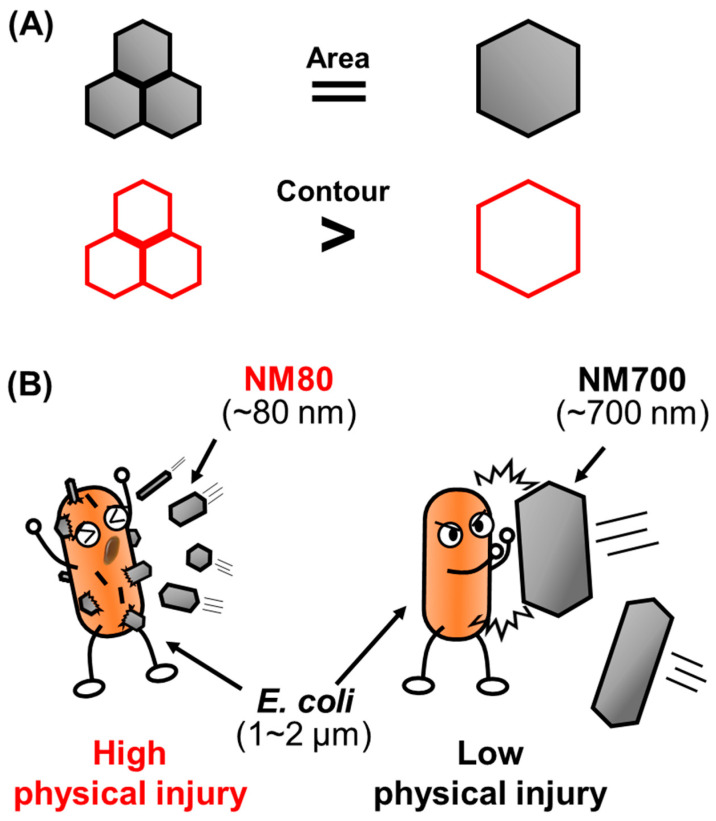
Illustration of the plausible mechanism of action of magnesium hydroxide nanoparticles against *E. coli*. (**A**) Magnesium hydroxide nanoparticles are assumed to be regular hexagons. Comparison of the contour of several smaller sizes with that of one large particle with the same area; (top, gray); the total contour length will be longer for smaller areas (bottom, red). (**B**) Image showing a comparison of the physical killing of *E. coli* by NM80 and NM700.

**Table 1 nanomaterials-11-01584-t001:** Bacterial strains used in this study.

Strains	Features	Reference
*E. coli* BW25113	*rrnB3* Δ*lacZ4787 hsdR514* Δ(*araBAD*)*567* Δ(*rhaBAD*)*568 rph*-*1*	[34]
*E. coli* BW25113 Δ*corA*	Δ*corA*, Km^R^	[34]
*E. coli* BW25113 Δ*mgtA*	Δ*mgtA*, Km^R^	[34]
*E. coli* BW25113 Δ*pitA*	Δ*pitA*, Km^R^	[34]
*E. coli* BW25113 Δ*pitB*	Δ*pitB*, Km^R^	[34]
*E. coli* BW25113 Δ*yifB*	Δ*yifB*, Km^R^	[34]

Km^R^ indicates kanamycin resistance.

**Table 2 nanomaterials-11-01584-t002:** Characterization data of NM80, NM300, and NM700 obtained using DLS, the laser diffraction particle size analyzer, ICP, and the pH meter. The analyses were performed at a concentration of 500 mg/L.

	Size (nm)			Mg^2+^		
	D_10_	D_50_	D_90_	(ppm)	pH	(°C)
NM80	54.3	75.2	118.8	5.78	10.32	23.7
NM300	206	328	626	4.85	10.26	23.5
NM700	443	726	1494	5.20	10.21	23.8

## Data Availability

The data presented in this article is available upon request from the corresponding author.

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
