# Peer review of "Magnesium Hydroxide Nanoparticles Kill Exponentially Growing and Persister Escherichia coli Cells by Causing Physical Damage"

_nanomaterials, 2021, doi:10.3390/nano11061584_

Round 1
Reviewer 1 Report
1.magnesium hydroxide nanoparticles purchased or syntheized ,icase synthesis add the experiments and results and discussion ,incase purchased mention the clear details
2. section 2.3 how much concentrations used of magnesium hydroxide nanoparticles
3.in XRD add the JCPDS number
4.authour why not studied the FTIR,UV and TEM analysis of nanoparticles
5.in SEM image the scale bar measurement not properly visible mention the scale bar and magnification in the text or figure
6.what is the MIC and MBC value of the nanoparticles
7.some of the place E.Coli not italics check and revise
8.stastical analysis missing
Author Response
Thank you very much for your helpful comments.
- The magnesium hydroxide nanoparticles used in this study were synthesized in our laboratory. As suggested, we have added the method of their synthesis in the Materials and Methods section (2.1, line 61-72).
- We used magnesium hydroxide nanoparticles at a concentration of 10 mg/l to 500 mg/l in this study, and this explanation has been added in the main text (section 2.3, line 97).
- As suggested, we added the ICDD number for XRD analysis (section 2.1, line 83).
- FTIR uses infrared light to analyze the presence of molecular bonds. However, in our study, there is no need to check for inorganic materials alone. We assume you are suggesting that nanoparticles will exhibit a high UV permeability. Particles of size less than 10 nm may have an effect on transmittance. The nanoparticles in this study are approximately 70 nm in size, so we concluded that there was no need for UV analysis. Although TEM allows observation at a higher magnification than SEM, we did not perform TEM analysis on our sample because the particle shape could be confirmed sufficiently with SEM.
- We revised the scale bars in Figure 5 and added the magnification details in the legends of Figures 1 and 5.
- Both MIC and MBC values of NM80, NM300, and NM700 were >1,000 mg/l. A value more than 1,000 mg/l causes a certain level of sedimentation, posing challenges to analysis.
- As you correctly pointed out, the species name was not in italics. We have corrected this error (section 3.3, line 175).
- As suggested, we have now added the details of the statistical analysis in the legends for Figures 2, 3, 4, and 6.
Reviewer 2 Report
The manuscript by Y. Nakamuraet al., entitled “Magnesium hydroxide nanoparticles kill Escherichia coli exponential or persister cells physically” presents a study about the influence of the size of magnesium hydroxide nanoparticles in their antibacterial activity. Nevertheless, some points are not sufficiently discussed and should be improved:
-. The authors do not provide how the nanoparticles were obtained. Are the nanoparticles synthesized by the authors or purchased? In any case the synthesis method should be commented and detailed.
-. The Scherrer equation is used to calculate the crystallite size in each sample, but the sentence used to relate it to nanoparticle size is unclear and insufficient.
-. The concentration-dependent bactericidal effect is only presented for NM80 sample. For the presented conclusions it would be relevant to compare that behaviour in all samples.
-. What is the reason for using MgSO4 at the specific concentration of 500 mg/l and no other as a control when comparing the antibacterial activity of the tree samples?
-. The authors addressed the effect of Mg ions on bacterial death to conclude that Mg ions have no effect on antibacterial activity. To validate this conclusion a study of ion release is needed.
-. The higher antibacterial effect of smaller nanoparticles (NM80 sample) is attributed to nanoparticles contour. Obviously, the nanoparticle size/bacteria size ratio (nanoparticle contour/bacteria size) is involved and not only the nanoparticle contour.
Other minor questions:
-. Line 108, “and MN had had the largest…” should be “and MN700 had had the largest…”
-. mg/L should be mg/l in all throughout the manuscript.
-. According to the figure 6 caption, there’s a mistake in the legend “PBS (persister)”.
For these reasons, I consider that the manuscript couldn’t be accept in this journal.
Author Response
Thank you very much for your helpful comments.
- The magnesium hydroxide nanoparticles used in this study were synthesized in our laboratory. As suggested, we added the method of their synthesis in the Materials and Methods section (2.1, line 61-72).
- As suggested, we have revised the text regarding XRD analysis using Scherrer equation. This explanation was added in section 3.1 (line 126), and the XRD data is shown in Table S1.
- As suggested, we performed experiments at different concentrations (10, 50, and 100 mg/l) of NM300 and NM700. We observed no bactericidal effect for NM300 and NM700. We have added these results as Figure S2 and described them in the main text (section 3.2, line 155).
- The magnesium hydroxide nanoparticles (NM80) showed a high sterilization effect at 500 mg/l. To compare the bactericidal effect of magnesium ions and the nanoparticles, the same concentration of MgSO4 was used. As discussed in the Discussion section, sulfates are commonly used for this purpose.
- CorA and MgtA are also responsible for the discharge of magnesium ions. Strains with mutations in these genes were not shown to be inhibited by the nanoparticles, as measured by the survival ratio (Figure 4). Therefore, we conclude that magnesium ions have no effect on the antibacterial activity.
- Yes, we agree with your comment. To clarify the ratio relationship between the size of the nanoparticles and the size of the sterilization target, it is necessary to verify these results with larger microorganisms (e.g., fungi and protozoa). We added this explanation in the Discussion section (line 247).
- We have revised this sentence (line 121, previously line 108).
- As suggested, we revised “mg/L” to “mg/l” in the manuscript.
- As you pointed it out, we have revised the legend of Figure 6.
Round 2
Reviewer 2 Report
The authors have answered the questions and corrected the manuscript according to the suggestions. I consider now the article can be accepted for publication
Author Response
Thank you for your helpful comments and suggestions, which has helped us improve the quality of our manuscript. The manuscript has been proofread by a native English speaker, and moderate English language changes have been made. Therefore, I look forward to its publication in Nanomaterials.